# Combined Fluorescence Fluctuation and Spectrofluorometric Measurements Reveal a Red-Shifted, Near-IR Emissive Photo-Isomerized Form of Cyanine 5

**DOI:** 10.3390/ijms24031990

**Published:** 2023-01-19

**Authors:** Elin Sandberg, Joachim Piguet, Haichun Liu, Jerker Widengren

**Affiliations:** Experimental Biomolecular Physics, Department of Applied Physics, Royal Institute of Technology (KTH), Albanova University Center, 106 91 Stockholm, Sweden

**Keywords:** cyanine, isomerization, FCS, TRAST, Cy5, single molecule, emission red-shift, near-IR, photodynamics, excitation modulation

## Abstract

Cyanine fluorophores are extensively used in fluorescence spectroscopy and imaging. Upon continuous excitation, especially at excitation conditions used in single-molecule and super-resolution experiments, photo-isomerized states of cyanines easily reach population probabilities of around 50%. Still, effects of photo-isomerization are largely ignored in such experiments. Here, we studied the photo-isomerization of the pentamethine cyanine 5 (Cy5) by two similar, yet complementary means to follow fluorophore blinking dynamics: fluorescence correlation spectroscopy (FCS) and transient-state (TRAST) excitation–modulation spectroscopy. Additionally, we combined TRAST and spectrofluorimetry (spectral-TRAST), whereby the emission spectra of Cy5 were recorded upon different rectangular pulse-train excitations. We also developed a framework for analyzing transitions between multiple emissive states in FCS and TRAST experiments, how the brightness of the different states is weighted, and what initial conditions that apply. Our FCS, TRAST, and spectral-TRAST experiments showed significant differences in dark-state relaxation amplitudes for different spectral detection ranges, which we attribute to an additional red-shifted, emissive photo-isomerized state of Cy5, not previously considered in FCS and single-molecule experiments. The photo-isomerization kinetics of this state indicate that it is formed under moderate excitation conditions, and its population and emission may thus deserve also more general consideration in fluorescence imaging and spectroscopy experiments.

## 1. Introduction

Cyanine dyes are extensively used as probes in the life sciences but have over the years also found use as photosensitizers, mode-locking compounds and in solar cells [1]. Motivated by this manifold of applications, the photophysics of cyanine dyes have been extensively studied [2]. Using techniques, such as transient absorption spectroscopy [3,4,5,6,7,8,9,10,11], laser-induced optoacoustic [12,13], photo-stationary absorption and fluorescence experiments [11,14,15,16,17,18,19], and theoretical calculations of electronic-state energies and oscillator strengths [20,21], the main features of cyanine dyes have been well established; in room-temperature solutions, most ground-state cyanine dyes are in an all-*trans* (N) conformation. Following excitation into the excited singlet state, photo-isomerization typically takes place with a high quantum yield (Φ_iso_), while the fluorescence quantum yield (Φ_f_) is relatively low compared to other fluorophore labels. Intersystem crossing to a triplet state typically competes with photo-isomerization and takes place with a low quantum yield (Φ_isc_), which often can be disregarded. The photo-isomerization is highly dependent on the local environment and is reduced by increased viscosity [3], increased viscous drag by head group substituents [22], and steric constraints upon binding to, e.g., proteins [23]. However, despite extensive research, many photophysical mechanisms of cyanine dyes remain elusive.

For fluorescence applications within the life sciences, the use of cyanine dyes has in the past decades been boosted by the strong development of fluorescence-based single-molecule spectroscopy (SMS) and super-resolution microscopy (SRM) [24]. Here, properties, such as a large excitation cross section, high fluorescence brightness and photostability, and spectral compatibility to common laser excitation sources and single-photon-counting detectors, have made cyanine dyes a major fluorophore category of choice. Moreover, fluorophore blinking, caused by reversible transitions into long-lived, non-fluorescent dark states, such as triplet, photo-ionized, and photo-isomerized states, is also of central importance. At typical excitation conditions useful for SMS [25], dark-state transitions can compromise molecular brightness and signal-to-background conditions, lead to precursor states of permanent photobleaching, and obscure observations of single-molecule dynamic events of interest, occurring at similar timescales. At the same time, however, blinking, or the switching on and off of fluorescence emitters, is also an absolute prerequisite for all forms of SRM [26]. Particularly, in single-molecule localization SRM [27], cyanine dyes are the overall fluorophores of choice because of their switching properties. Yet, several questions remain open regarding underlying mechanisms [28,29] and on how to optimize these for SRM applications (see [30] for a review).

From the notion that fluorophore blinking properties are central in SMS, SRM, and essentially any application requiring high detection sensitivity, fast readout, or high resolution, we revisited the blinking kinetics of the widely used carbocyanine dye cyanine 5 (Cy5) using a combination of fluorescence correlation spectroscopy (FCS), transient-state (TRAST) spectroscopy, and spectrofluorometric and time-correlated single-photon counting (TCSPC) fluorescence decay measurements. Using FCS, single-molecule blinking due to dark-state transitions within fluorophores can be analyzed in a straightforward manner, typically seen as superimposed fluorescence fluctuations onto those due to translational diffusion of fluorescent molecules into and out of an open confocal detection volume [31]. Such FCS studies also include investigations of the photo-isomerization of Cy5 [32].

In TRAST [33,34,35,36,37,38], reversible transitions of the long-lived dark states of fluorescent molecules are characterized from how the time-averaged fluorescence intensity from the fluorophores varies with the modulation of the laser excitation intensity, with the modulation systematically varied over the timescales of the fluorophore dark-state kinetics. TRAST bears similarities to FCS but views blinking events differently, as the fluorescence response to an excitation pulse train instead of spontaneous single-molecule blinking events. In contrast to FCS, TRAST does not rely on single-molecule detection conditions or high time resolution, can be run with lower excitation intensities, and can typically be applied on a broader range of samples.

In this work, from a combination of FCS and TRAST experiments, offering similar, yet complementary means to follow fluorophore blinking, we find evidence for a second emissive photo-isomerized state of Cy5, with its emission red-shifted compared to that of the all-*trans* conformation. This finding was further supported with fluorescence decay measurements using TCSPC and using a combination of TRAST and spectrofluorimetry (spectral-TRAST), whereby the fluorescence emission spectra of Cy5 were recorded upon different rectangular pulse-train excitations. Using the spectral-TRAST approach, we could observe a clear relative increase in the longer-wavelength range of the fluorescence spectrum of Cy5 for longer excitation pulse durations, consistent with the formation of an emissive double-isomerized form of Cy5. Finally, we show how this additional emissive state can be included into models and analyses of FCS and TRAST data and discuss further implications in SMS, SRM, and fluorescence spectroscopy and imaging more generally.

## 2. Results

### 2.1. FCS and TRAST Experiments

FCS curves of Cy5 were recorded in an aqueous solution (12 mM PBS, pH 7.2) at 638 nm excitation using three different emission filters (Figure 1). Each of the FCS curves displayed a prominent dark-state relaxation, which could be well fitted to a simple two-state *trans-cis* isomerization model (Equation (8)). However, it was also found that the isomerization amplitudes, $A_{iso}$, varied significantly with the emission filter used, under otherwise identical experimental conditions. With an emission filter covering the full range of the Cy5 emission spectrum (645–795 nm), $A_{iso}$ was determined to 0.54, which is similar to what has previously been found under comparable experimental conditions [32]. With an emission filter covering the lower, blue-shifted part of the Cy5 emission spectrum (645–680 nm, hereinafter referred to as the B-filter) $A_{iso}$ was increased to 0.61. With a red-shifted emission spectrum (690–750 nm, hereinafter referred to as the R-filter), $A_{iso}$ instead significantly reduced (0.42). The use of different excitation wavelengths can be expected to change $A_{iso}$ in cyanine dyes described by the two-state isomerization model of Equation (8), with transitions between a fluorescent all-*trans* and a non-fluorescent *cis* conformation [32], as also found in experiments here, using a 660 nm instead of a 638 nm excitation laser (Appendix A). However, based on this model, $A_{iso}$ is not expected to change with the spectral range of emission detected (Figure 1). This rather indicates the presence of an additional emissive state of Cy5, with red-shifted emission compared to that of its all-*trans* conformation. To obtain further evidence for such a state, we performed TCSPC fluorescence decay measurements of Cy5 in the same PBS solution using pulsed 638 nm excitation (Appendix A). Whereas the fluorescence decay measured with the B-filter could be well fitted to a mono-exponential decay ($\tau_{f}$ = 1.0 ns), a bi-exponential decay model clearly generated a better fit when the R-filter was used ($\tau_{f}$ = 1.0 ns, $\tau_{f2}$ = 0.5 ns, with relative amplitudes of 0.62 and 0.38, respectively). This supports the presence of an additional emissive state, with a shorter lifetime than for the all-*trans* conformation, contributing to the fluorescence in the red-shifted part of the spectrum.

Next, we recorded FCS curves using either the blue-shifted(B-) or the red-shifted(R-) emission band pass filters and at different $\Phi_{exc}$ applied (Figure 2). In the FCS curves, for both filters used, the overall dark-state relaxation time decreased with increasing $\Phi_{exc}$. The amplitudes were almost independent of $\Phi_{exc}$ but clearly different depending on the emission filter used. As a complement to these FCS experiments, we also performed TRAST measurements on the same Cy5 sample at 638 nm excitation and using the same B- and R-filters (Figure 3). As for the FCS curves in Figure 2, the TRAST decay amplitudes were lower when the R-filter was used, but otherwise showed relaxations with a similar $\Phi_{exc}$ dependence as in the FCS curves, consistent with reversibly excitation-driven dark-state transitions. The decay amplitudes slightly decreased with lower $\Phi_{exc}$. This reflects that at the lower $\Phi_{exc}$ levels applied in the TRAST measurements, thermal back-isomerization rates are no longer negligible compared to the excitation-driven back-isomerization rates. Similar lowered populations of photo-isomerized states in Cy5 have also been observed in FCS experiments performed at lower $\Phi_{exc}$ [32].

### 2.2. Photophysical Model Based on the FCS and TRAST Experiments

Next, we wanted to bring the observations from the FCS, TRAST, and TCSPC measurements described before into a photophysical model. The photophysics of cyanine dyes have been extensively studied, and with *trans*-*cis* photo-isomerization reported as a dominating process [1,2,3,4,5,6,7,8,9,10,11,12,13,14,15,16,17,18,19,20,21]. The $\Phi_{exc}$ dependence observed in the FCS and TRAST experiments (Figure 2 and Figure 3) is also well in agreement with such reversibly excitation-driven *trans*-*cis* photo-isomerization, often described in terms of a potential energy surface model, including an all-*trans* and a mono-*cis* conformation (the Rulliére model) [19]. However, mono-*cis* conformations of cyanine dyes have typically been found to exhibit low fluorescence [3,12,13,14,24]. Thus, if not from a mono-*cis* conformation, the red-shifted emission observed in our experiments then should originate from an additional photo-isomerized state. A possible model, explaining the observations in the FCS, TRAST, and TCSPC measurements, is depicted in Figure 4A, which in turn can be reduced to the three-state model of Figure 4B. In addition to an all-*trans* form of Cy5 (N), and a photo-isomerized mono-*cis* form (P_1_), this model also includes an emissive, double-cis conformation (P_2_), which is formed from P_1_. Such a P_2_ state, formed with a two-step photo-induced process, is not typically found in (one-step) flash photolysis or transient absorption experiments but can be identified with, and has been inferred from, photokinetic steady-state fluorescence and absorption measurements [18]. In such a model, Cy5 can, in principle, reversely photo-isomerize between the N and P_2_ states via two (mono-)*cis* states. For simplicity, however, we represent these two states as one mono-*cis* state in our model. The resulting three-state model of Figure 4B was then used as a basis for fitting the recorded FCS and TRAST curves (see the caption for a description and definition of fitted parameters). The effective rates and the differential equations governing the population of the N, P_1_, and P_2_ states are described in Appendix A.

### 2.3. Experimental FCS and TRAST Data Fitted to the Photophysical Model

Based on the photophysical three-state model of Figure 4B and using Equations (5), (6), and (10) from the Materials and Methods section, the two sets of FCS curves in Figure 2 were jointly fitted in a global manner. For the FCS curves measured with the B-filter, we note that the Cy5 fluorescence decay in this emission range was found to be mono-exponential (Appendix A). This indicates that these FCS curves are generated from fluorescence originating from N only. In the fit, we could thus fix the relative brightness of P_2_, *Q*, to 0. Likewise, with *Q* = 0, emission will only be detected following decay to the ground state of N. It should be borne in mind that FCS curves generally reflect the probability to detect a fluorescence photon from a molecule at correlation time, τ, given that a fluorescence photon was detected from the same molecule at τ = 0. The initial condition thus should reflect the probabilities of detecting a fluorescence photon from any of the states N, $P_{1}$, and $P_{2}.$ For the B-filter curves, *Q* = 0, and fluorescence photons can thus only be detected from N, and the initial condition is $\left[\left[N\right]\left(\bar{r},0\right),\;\left[P_{1}\right]\left(\bar{r},0\right),\;\left[P_{2}\right]\left(\bar{r},0\right)\right]=\left[1,\;0,\;0\;\right]$, as stated in Equation (S5). However, for FCS curves recorded with the R-filter, different initial conditions for the state populations across the FCS detection volume $\left[N\right]\left(\bar{r},t\right),\;\left[P_{1}\right]\left(\bar{r},t\right)$ and $\left[P_{2}\right]\left(\bar{r},t\right)$ apply. For FCS curves recorded with the R-filter, $P_{2}$ has a relative brightness of *Q* compared to N. The initial condition then reflects the probabilities that a detected fluorescence photon comes from N or P_2_ and can be approximated by:(1)$$\left[N\right]\left(\bar{r},0\right)=\frac{\bar{N}\left(\bar{r}\right)}{\bar{N}\left(\bar{r}\right)+Q{\bar{P}}_{2}^{}\left(\bar{r}\right)}\;\left[P_{2}\;\right]\left(\bar{r},0\right)=\frac{Q{\bar{P}}_{2}^{}\left(\bar{r}\right)}{\bar{N}\left(\bar{r}\right)+Q{\bar{P}}_{2}^{}\left(\bar{r}\right)}\;\left[P_{1}\;\right]\left(\bar{r},0\right)=0$$
where $\bar{N}\left(\bar{r}\right)$ and ${\bar{P}}_{2}^{}\left(\bar{r}\right)$ denote the steady-state populations of the N and P_2_ states, respectively, at a location $\bar{r}$ in the FCS detection volume. In the fitting of the FCS curves in Figure 2, we thus used the initial condition of Equation (1) as the initial condition for the state populations across the FCS detection volume for the R-filter curves and Equation (S5) for the B-filter curves. Since *Q* is included in the initial condition, *Q* was fitted globally (to the R-filter curves) in an iterative manner, with the initial condition of Equation (1) updated with each iteration until a stable *Q*-value was reached. For all FCS curves, given the relatively high $\Phi_{exc}$ levels applied in the FCS experiments, the contribution from the thermal back-isomerization rates, $k_{biso1}^{Th}$ and $k_{biso2}^{Th}$, could be neglected compared to the excitation-driven contributions to the effective back-isomerization rates, $k_{biso1}´$ and $k_{biso2}´$. $k_{biso1}^{Th}$ and $k_{biso2}^{Th}$ could thus be fixed to 0 in the fit. Finally, in the fit, $\sigma_{N}$ was fixed to 6.2$\times 10^{-16}$ cm^2^ (from [32], scaled by the absorption spectrum to obtain $\sigma_{N}$ at 638 nm) and $\tau_{F}$ to 1.0 ns, as obtained with TCSPC measurements (Appendix A). With these prerequisites, initial conditions, and fixed parameter values, global fitting based on Equation (10), and including both sets of FCS curves in Figure 2, could then successfully reproduce the experimental curves and yielded the following parameter values: $k_{iso}$ = 36 µs^−1^, $\sigma_{biso1}=0.25\times 10^{-16}$ cm^2^, $\sigma_{iso2}=0.31\times 10^{-16}$ cm^2^, $\sigma_{biso2}=0.25\times 10^{-16}$ cm^2^, and Q=0.46.

Next, both sets of TRAST curves (Figure 3), recorded with the B- and R-filters, respectively, were likewise jointly fitted with global parameters. As in the fit of the FCS data, $\sigma_{N}$ was fixed to 6.$2\times 10^{-16}$ cm^2^ and $k_{10}^{N}$ = 1/$\tau_{f}$ − $k_{iso}$, with $\tau_{f}$ fixed to 1.0 ns. For the TRAST curves recorded with the B-filter, *Q* was fixed to 0 (for the same reasons as given before for the FCS fitting), while for the curves recorded with the R-filter, *Q* was fitted globally. For the fitting of the TRAST curves, and irrespective of the emission filter used, the same initial condition for the state populations could be applied (Equation (S5)). The fitting (to Equations (2)–(4)) resulted in curves that could successfully reproduce the experimental TRAST curves and yielded the following parameter values: $k_{iso}$ = 33 µs^−1^, $\sigma_{biso1}=0.26\times 10^{-16}$ cm^2^, $\sigma_{iso2}=0.3\times 10^{-16}$ cm^2^, $\sigma_{biso2}=0.25\times 10^{-16}$ cm^2^, $k_{th1}$ = 0.015 µs^−1^, $k_{th2}$ = 0.08 µs^−1^, and Q = 0.28 (for the TRAST curves recorded with the R-filter). The fitted parameter values are also in good agreement between the FCS and TRAST data. While Q can be different in the FCS and TRAST experiments, due to possible differences in detection quantum yields between the setups used (Equation (2)), the fitted values indicate that such differences are small. It can be noted that the overall isomerization amplitudes show up differently in FCS than in TRAST experiments and with a larger contrast between the amplitudes in the FCS curves for the different emission filters used. This follows from the different initial conditions that apply in FCS versus TRAST experiments, the different weighting (by *Q*^2^ for the FCS curves (Equation (10)), by *Q* only for the TRAST curves (Equations (2)–(4)), and different $\Phi_{exc}$ levels applied in FCS and TRAST experiments.

From this fitting outcome, we can conclude that the photophysical kinetics reflected in the experimental FCS and TRAST curves are quite compatible with the three-state photo-isomerization model of Figure 4B and the formation of a second red-shifted emissive state in Cy5 upon excitation. Figure 4C shows the calculations of how the N, P_1_, and P_2_ state populations of Cy5 can evolve upon excitation, based on the fitted parameter values, and considering different excitation intensities applied, further illustrating how the underlying state population kinetics contribute to the observed relaxations in the TRAST experiments. Figure 4D shows how the N, P_1_, and P_2_ state populations depend on $\Phi_{exc}$ at CW excitation. It can be noted that for $\Phi_{exc}$ < 5 kW/cm^2^, the populations are clearly $\Phi_{exc}\;$ dependent, while for a higher $\Phi_{exc},$ no major effects on the steady-state are found. At $\Phi_{exc}$ approaching 100 kW/cm^2^, however, excited-state saturation effects set in, particularly in N (having the longest excited-state lifetime).

### 2.4. Spectral-TRAST Experiments

To obtain further evidence for the formation of a second red-shifted, emissive photo-isomerized state, we established a spectrofluorimetric procedure using rectangular pulse-train excitation, with the excitation modulation varied systematically, in a similar manner as in the TRAST experiments described before. The results of such spectral-TRAST experiments are shown in Figure 5A–C. Indeed, at constant $\Phi_{exc}$ within the pulses, and with increased pulse durations, *w* (keeping the excitation pulse-train duty cycle the same), the recorded fluorescence emission spectra (Figure 5A) displayed a prominent decrease at the emission peak wavelength (~670 nm) and a more minor decrease at longer wavelengths. When normalizing the emission spectra to unity at the peak wavelength (Figure 5B), we thus observed a clear relative red-shift in the recorded spectra with longer *w*, consistent with the formation of a red-shifted, emissive photo-isomerized state. Plotting the fluorescence intensity within different spectral bands of the spectra as a function of *w* (Figure 5C) then allowed us to generate TRAST curves for different emission ranges over the full emission spectrum of Cy5. In these TRAST curves, we observed the same trends as in the previous TRAST and FCS experiments—a clear decrease in the relaxation amplitudes for longer emission wavelengths, and well in agreement with the hypothesized model (Figure 4B).

## 3. Discussion

A combination of FCS, TRAST, spectral-TRAST, and TCSPC measurements suggests that a red-emissive photo-isomerized state can be formed in Cy5 upon excitation. The experimental findings are well in agreement with the formation of a dark photo-isomerized state, P_1_, and then from P_1_ an additional, double-photo-isomerized state, P_2_, with both its excitation and emission spectra red-shifted compared to its all-*trans* state, N. Since P_2_ is generated with a two-step photo-induced process, it is not expected to be clearly seen and has generally also not been reported from (one-step) flash photolysis or transient absorption experiments of pentamethine cyanine dyes. However, it has been implicated from a few photokinetic steady-state fluorescence and absorption studies [18]. From our experimental data and their analyses, we cannot fully exclude other photokinetic models of Cy5 than the three-state isomerization model of Figure 4B. Delayed fluorescence following triplet-state formation has been reported from Cy5 in deoxygenated ethanol solutions [9,39]. However, triplet states are effectively quenched with molecular oxygen under our experimental conditions, in air-saturated aqueous solutions. Moreover, the intersystem crossing yields of Cy5 are more than an order of magnitude lower than for isomerization and back-isomerization [32], and triplet state formation can thus be disregarded in our model. However, it is also possible to fit the experimental FCS and TRAST curves with a two-state isomerization model (Appendix A) instead of the three-state model of Figure 4B. In such two-state model, with P_1_ and P_2_ merged into one photo-isomerized state, P, the merged P state may either indeed represent a single weakly fluorescent mono-*cis* state or a time average of one or several photo-isomerized forms and a double-photo-isomerized form of Cy5 (red-marked in Figure 4B). However, given that photo-isomerized, mono-*cis* conformations of pentamethine cyanine dyes typically have been found to exhibit low fluorescence [3,12,13,14,24], the three-state model of Figure 4B is the simplest model to incorporate such features. Moreover, irrespective of the underlying mechanisms and model, the implications of the experimental findings are in several aspects the same. Our data (Figure 4D) show that the formation of a red-shifted, emissive photo-isomerized state of Cy5 can take place at quite moderate excitation intensities (below kW/cm^2^), with a concomitant, red-shifted emission spectrum. This may have to be accounted for in multi-color imaging and spectroscopy experiments with Cy5, based on, e.g., linear unmixing, as well as in Förster resonance energy transfer (FRET) experiments, where the spectral shape of Cy5 also matters for a correct interpretation of the data. The added knowledge of Cy5 photo-isomerization provided in this work can also be useful for optimization of excitation conditions in SMS and SRM experiments. Moreover, knowledge of the photodynamics of Cy5, as presented in this work, can be used to design excitation strategies to specifically promote the formation of more red-shifted emission in Cy5. Cy5 with a more red-shifted, NIR emission may enable fluorescence measurements with lowered background and scattering, or the use of Cy5 as a NIR photosensitizer. From a methodological point of view, the parallel use of TRAST and FCS in this work makes it, on the one hand, possible to map the photodynamics of Cy5 with two independent techniques and, on the other hand, also to obtain complementary data. Particularly, with TRAST, lower excitation intensities can be applied, slower isomerization relaxation times can be studied (beyond the typical passage times of Cy5 through the FCS detection volume), and thermal back-isomerization rates can be determined. We showed how the relative brightness of the generated photo-induced states and the initial conditions for their populations come differently into play in FCS and TRAST analyses, also depending on the experimental conditions. This provides a framework for analyses of FCS and TRAST measurements, including photo-induced emissive states kinetics, in cyanine dyes as well as in other fluorophores. The fact that the FCS and TRAST curves could be fitted to the same photophysical model, and that fitting generated similar fitted parameters, gives further support to the experiments and these analyses.

In conclusion, using a combination of FCS and TRAST experiments we provided evidence for a second emissive photo-isomerized state of Cy5, with a red-shifted emission compared to that of its all-*trans* conformation. Additional evidence was provided by combining TRAST and spectrofluorimetry (spectral-TRAST), showing significant differences in fluorescence emission spectra upon different rectangular pulse-train excitations. Both FCS and TRAST data could be well fitted to a photo-isomerization model, yielding similar fitted parameter values, with the isomerization rate, $k_{iso}$, determined to 36 µs^−1^ and 33 µs^−1^, respectively. This is close to previously reported FCS data (with $k_{iso}$ reported to be 25 µs^−1^ [32]). As shown in this work, the lower the relative brightness difference between the switched states, the smaller the relaxation amplitudes in experimental FCS and TRAST curves. The formation rates and populations of partly emissive states will thus be underestimated in FCS and TRAST experiments if they are considered fully non-emissive. Additionally, the emissive photo-isomerized state, as found for Cy5 in this work, likely also needs to be taken into consideration in a broad range of SMS, SRM, FRET, and multi-color experiments in which Cy5 is used.

## 4. Materials and Methods

### 4.1. Sample Preparation

Stock solutions of sulfonated cyanine5 NHS ester (Cy5; Lumiprobe GmbH, Hannover, Germany) were prepared in phosphate buffered saline (12 mM, pH 7.2), stored in the fridge at 8 °C, and then diluted with the same solvent just before measurements to the desired concentration.

### 4.2. TRAST Experiments

TRAST measurements were carried out on a home-built TRAST setup based on an inverted epi-fluorescence microscope (Olympus, Tokyo, Japan, IX70) and modified from a previously described arrangement [33]. In short, fluorescence was excited with a beam of a diode laser (638 nm, 200 mW, Cobolt, Solna, Sweden) passing an excitation filter (Semrock BrightLine 637/7) and an acousto-optic modulator (AOM; AA Opto Electronics, Orsay, France, MQ180-A0,25-VIS) and then being reflected by a dichroic mirror (Chroma, ZT640rdc) and focused close to the back aperture of the objective (Olympus, Tokyo, Japan, UPLSAPO 60×/1.20 W) to produce a wide-field illumination in the sample (beam waist *ω*_0_ = 20 µm (1/e^2^ radius)). The fluorescence signal was collected with the same objective, passed through the same dichroic mirror and an emission filter (HQ720/150, Chroma, 680 nm blocking edge Brightline, Semrock, in combination with the last-mentioned filter or 710/40 Brightline, Semrock), and then fed to an sCMOS camera (Hamamatsu ORCA-Flash4.0 V3). The experiments were controlled and synchronized with custom software implemented in Matlab. A digital I/O card (PCI-6602, National Instruments) was used to trigger the camera and generate random excitation pulse trains sent to the AOM driver unit. For the spectral experiments, a function generator was used for AOM triggering and generation of the different pulse durations. The fluorescence signal was passed through an aperture (centered around the emission intensity maximum and with 23% of the fluorescence passing through, ensuring that only fluorescence from the center of the excitation volume is detected by the camera), passed through an emission filter (647 nm RazorEdge, Semrock ultrasteep long-pass-edge filter), and then focused into the multimode fiber (diameter 600 µm) of a fiber-coupled spectrometer (QEPro, Ocean Optics). The obtained emission spectra were corrected by normalizing them with the wavelength detection efficiency function and the grating efficiency for different wavelengths, since this is not completely uniform over the spectral width of the Cy5 emission spectrum.

### 4.3. TRAST Analysis

To calculate the recorded fluorescence intensity in the TRAST experiments, we used the reduced photophysical model for Cy5 in Figure 4B, with two emissive states, the all-*trans* state, N, and the double-*cis* state, P_2_, and with the two (mono-)*cis* states of Cy5 non-luminescent, and for simplicity represented as one state, P_1_. For a homogeneous solution sample, and from the rate equations of a Cy5 fluorophore subject to a rectangular excitation pulse starting at t = 0 (Appendix A), the fluorescence signal recorded in our experimental setup can be described by
(2)Ft=c·q1F·q1D·σN∭CEFr¯·Φexcr¯·Nr¯,t+Q·P2r¯,t dV

Here, $\left[N\right]$ and $\left[P_{2}\right]$ denote the probabilities that each of these emissive states (in either their ground or their excited states) are populated in the fluorophores, and Q=(q2F·q2D·σP2)/(q1F·q1D·σN) is the relative brightness of P_2_ compared to N, where q1D and q2D denote the overall detection quantum yield of the emission from the excited singlet state of N and P_2_, respectively, and q1F and q2F are the fluorescence quantum yields of these states. $\sigma_{N}$ and $\sigma_{P_{2}}$ denote the excitation cross section of the ground singlet state of N and P, respectively; $CEF\left(\bar{r}\right)$ is the collection efficiency function of the detection system; and c is the fluorophore concentration.

At onset of excitation, $F\left(t\right)$ will show characteristic relaxation on a μs-to-ms timescale, reflecting changes in the population of the emissive state(s); see Appendix A). Similar relaxations can also be observed in the time-averaged fluorescence signal resulting from a rectangular excitation pulse of duration w
(3)$$\langle F_{exc}\left(w\right)\rangle =\frac{1}{w}\int_{0}^{w}F\left(t\right)\;dt$$
when w is increased from the μs-to-ms time range. Analyzing how $\langle F_{exc}\left(w\right)\rangle$ varies with w then allows the population kinetics of long-lived photo-induced states of the fluorophore to be determined, which is the general basis for TRAST monitoring.

To obtain sufficient photon counts, even for short w, we collected the total signal resulting from an excitation pulse train of *N* identical pulse repetitions. *N* was adjusted to maintain a constant laser illumination time, $t_{ill}=N\cdot w$, for all w. The illumination-time was varied between 1 and 10 ms depending on the emission count rate. A so-called TRAST curve was then produced by calculating the time-averaged fluorescence signal during excitation for each pulse train, normalized for a given pulse duration $w_{0}$
(4)$$\begin{array}{c} \langle F_{exc}{\left(w\right)\rangle }_{norm}=\left(\frac{1}{N}\overset{N}{\underset{i=1}{\sum }}\langle F_{exc}{\left(w\right)\rangle }_{i}\right)/\left(\frac{1}{N_{0}}\overset{N_{0}}{\underset{i=1}{\sum }}\langle F_{exc}{\left(w_{0}\right)\rangle }_{i}\right)\;\; \end{array}$$

The pulse duration used for normalization, $w_{0}$, was chosen to be short enough (typically sub-μs) not to lead to any noticeable build-up of dark transient states, yet longer than the anti-bunching rise time of $F\left(t\right)$ upon onset of excitation, which typically is in the nanosecond time range [40].

In the above expression, $\langle F_{exc}{\left(w\right)\rangle }_{i}$ represents the total signal collected from the i-th pulse in the pulse train, as defined in Equation (3). By using a low-excitation duty cycle, here η=0.01−0.001, fluorophores are allowed to fully recover back to S_0_ before the onset of the next pulse. In the normalization step of Equation (4), several parameters used to calculate $F\left(t\right)$ in Equation (2) cancel out. The final expression for $\langle F_{exc}{\left(w\right)\rangle }_{norm}$ therefore becomes independent of c, as well as of the absolute $q_{D}$ and $q_{F}$ values for the two emissive species.

A complete TRAST experiment consisted of a stack of 30 fluorescence images. Each image represents the total fluorescence signal from an entire excitation pulse train, captured using a camera exposure time of $t_{exp}=t_{ill}/\eta$. Pulse durations, w, were distributed logarithmically between either 100 ns and 1 ms or 100 ns and 10 ms. They were measured in randomized order to avoid bias due to time effects. An additional 10 reference frames, all using a 100 ns pulse duration to avoid dark-state build-up, were inserted at regular intervals between the 30 main images to track any permanent bleaching of the sample.

The TRAST data were analyzed using software implemented in Matlab, as previously described [33,36,37]. The recorded TRAST data were first pre-processed with subtraction of the static ambient background, optional binning to either larger pixels or regions of interest (ROIs) within the recorded images, and correction for bleaching (see TRAST spectroscopy before). The overall bleaching was maximally 5–10% of the total detected intensity.

In all measurements, TRAST curves were produced by calculating $\langle F_{exc}{\left(w\right)\rangle }_{norm}$ within a ROI corresponding to a 20 μm radius in the sample plane, centered on the excitation beam. The fitting of photophysical rate parameters was then performed by simulating theoretical TRAST curves using Equations (2)–(4) and comparing them to the experimental data. The set of rate parameter values best describing the experimental data was then found using non-linear least squares optimization. In the fit, the excited-state lifetime, $\tau_{f}$, of N was fixed to its fitted value, determined with time-correlated single photon counting measurements (see the Results section), with $1/\tau_{f}$ comprising all deactivation (including isomerization) rates from the excited states of N. Relative differences in the excitation rates between N and P_2_ are accounted for in the relative brightness parameter, *Q* (Equation (2)), but could only be indirectly determined for P_2_ (as part of a back-isomerization cross section, see the Results section). For N, an average singlet excitation rate, ${\hat{k}}_{01}$, was calculated for each ROI using Equation (S9), as previously described [33,36,41,42] (see also Appendix A for details) using an excitation cross section of $\sigma_{N}=6.2\times 10^{-16}\;{\mathrm{cm}}^{2}$ [32] (at 638 nm excitation).

### 4.4. FCS Experiments

FCS measurements were performed on a commercial, epi-illuminated, confocal laser scanning microscope (Olympus, Tokyo, Japan, FV1200). Cy5 in aqueous solution was excited with a focused beam (338 nm, 1/e^2^ radius) of a 638 nm laser (LDH-D-C-640 from PicoQuant GmbH, Berlin, Germany) in continuous wave. The emitted fluorescence was collected back through the microscope objective (UPlanSApo 60x/1.2 w, Olympus, Tokyo, Japan), passed through a dichroic mirror (ZT405/488/635rpc-UF2, Chroma), an emission filter (HQ720/150, Chroma, 680 nm blocking edge Brightline, Semrock, in combination with last mentioned filter or 710/40 Brightline, Semrock), and focused onto a pinhole (50 µm diameter) in the back focal plane. The fluorescence signal was finally split and directed on two avalanche photodiodes (Tau-SPAD, PicoQuant GmbH, Berlin, Germany), whose signals were collected with a data acquisition card (Hydraharp 400, Picoquant, Berlin, Germany).

### 4.5. FCS Analysis

In the FCS measurements, for freely diffusing fluorescent molecules undergoing dark-state transitions, the autocorrelation curves of the recorded fluorescence intensity, $F\left(t\right)$, can be described by:(5)$$G\left(\tau \right)=\frac{<F\left(t\right)F\left(t+\tau \right)>}{<F\left(t\right){>}^{2}}=G_{D}\left(\tau \right)G_{T}\left(\tau \right)+1$$
where $G_{D}\left(\tau \right)$ denotes the translational diffusion-dependent part and $G_{T}\left(\tau \right)$ signifies the contribution from photo-induced dark state transitions. $G_{D}\left(\tau \right)$ can be expressed as:(6)$$G_{D}\left(\tau \right)=\frac{1}{N_{m}}{\left[1+\frac{\tau }{\tau_{D}}\right]}^{-1}\times {\left[1+{\left(\frac{\omega_{0}}{\omega_{z}}\right)}^{2}\frac{\tau }{\tau_{D}}\right]}^{-\frac{1}{2}}$$
with $\omega_{0}$ and $\omega_{Z}$ denoting the distances from the center of the laser beam focus in the radial and axial direction, respectively, at which the collected fluorescence intensity has dropped to $1/e^{2}$ of its peak value. $N_{m}$ is the mean number of fluorescent molecules within the detection volume. $\tau_{D}$ is the characteristic diffusion time of the fluorescent molecules, given by the diffusion coefficient D as $\tau_{D}=\omega_{0}^{2}/4D$.

For a fluorophore with one emissive state (within which excitation/deexcitation cycles between a ground and an excited singlet state take place on a timescale much faster than the correlation times considered), and with no dark-state transitions, the blinking term in Equation (5), $G_{T}\left(\tau \right)=1$. Otherwise, with *n* dark transient states, and for τ much longer than the anti-bunching relaxation times of the fluorophores, $G_{T}\left(\tau \right)$ can be expressed as a normalized set of relaxation terms [31], averaged over the confocal detection volume, and weighted by the square of the detected molecular brightness of the molecules, $W\left(\bar{r}\right)$: (7)$$G_{T}\left(\tau \right)=\frac{\int W^{2}\left(\bar{r}\right)\left[1-\sum_{i=1}^{n}\left[A_{i}\left(\bar{r}\right)-A_{i}\left(\bar{r}\right)e^{-\lambda_{i}\left(\bar{r}\right)\tau }\right]\right]dV}{\int W^{2}\left(\bar{r}\right)\left[\sum_{i=1}^{n}\left[1-A_{i}\left(\bar{r}\right)\right]\right]dV}$$

Here, $W\left(\bar{r}\right)=CEF\left(\bar{r}\right)\sigma_{exc}\Phi_{exc}\left(\bar{r}\right)\Phi_{F}\Phi_{D}$, with $\sigma_{exc}$ denoting the excitation cross section of the emissive state. $\lambda_{i}\left(\bar{r}\right)$ are the eigenvalues and $A_{i}\left(\bar{r}\right)$ the related amplitudes, reflecting the population build-up of the different photo-induced non-fluorescent states. At steady-state and with no photobleaching, the sum of the population probabilities for S_0_ and S_1_, together with $\sum_{i=1}^{n}\left[A_{i}\left(\bar{r}\right)\right]$, equals 1.

For diffusing cyanine fluorophores, which can be considered to undergo *trans*-*cis* isomerization between two states only, a fluorescent *trans* (N) state and a dark mono-*cis* (P_1_) state, and assuming uniform excitation conditions within the FCS detection volume, the recorded autocorrelation curves (FCS curves) can be expressed as [32,43]
(8)$$G\left(\tau \right)=\frac{1}{\left(1-A_{iso}\right)}G_{D}\left(\tau \right)\left(1-A_{iso}+A_{iso}e^{-\frac{\tau }{\tau_{iso}}}\right)+1\;\;\;\;\;\;\;\;\;\;\;\;$$
where $A_{iso}{}_{}$ is the isomerization relaxation amplitude and, for the case of this two-state isomerization model, corresponds to the averaged steady-state fraction of P_1_ in the detection volume upon excitation, and $\tau_{iso}$ is the average isomerization relaxation time. For recorded FCS curves fitted to Equation (8),$\;N_{m}$, $A_{iso}$, $\tau_{iso}$, $\tau_{D},$ and $S=\omega_{z}/\omega_{0}$ were used as fitted parameters, with the fitting based on a non-linear least squares optimization routine written in Matlab.

For a cyanine fluorophore that apart from an all-*trans* state, N, can photo-isomerize into a non-fluorescent mono-*cis* (P_1_) and an emissive double-*cis* (P_2_) conformation, as described by the model of Figure 4B, Equation (7) changes into: (9)$$G_{T}\left(\tau \right)=\frac{\int W^{2}\left(\bar{r}\right)\left[\left[1-\sum_{i=2}^{3}\left[A_{i}\left(\bar{r}\right)-A_{i}\left(\bar{r}\right)e^{-\lambda_{i}\left(\bar{r}\right)\tau }\right]\right]+Q^{2}\left[1-\sum_{i=1}^{2}\left[A_{i}\left(\bar{r}\right)-A_{i}\left(\bar{r}\right)e^{-\lambda_{i}\left(\bar{r}\right)\tau }\right]\right]\right]dV}{\int W^{2}\left(\bar{r}\right)\left[\left[\sum_{i=2}^{3}\left[1-A_{i}\left(\bar{r}\right)\right]\right]+Q^{2}\left[\sum_{i=1}^{2}\left[1-A_{i}\left(\bar{r}\right)\right]\right]\right]dV}$$

Here, the amplitudes $A_{i}\left(\bar{r}\right)$ refer to the steady-state populations of N, P_1_, and P_2_ for *i* = 1, 2, and 3, respectively, and $\lambda_{i}\left(\bar{r}\right)$ are the corresponding relaxation rates/eigenvalues. $W\left(\bar{r}\right)$ refers to the molecular brightness of N, and *Q* is the relative brightness of P_2_ compared to N, as defined in Equation (2). With the population probabilities for N, P_1_, and P_2_ after onset of constant excitation, $\Phi_{exc}\left(\bar{r}\right),$ at time t=0, $\left[N\right]\left(\bar{r},t\right),\;\left[P_{1}\right]\left(\bar{r},t\right)$, and $\left[P_{2}\right]\left(\bar{r},t\right)$, defined as in Appendix A can be written as: (10)$$G_{T}\left(\tau \right)=\frac{\int W^{2}\left(\bar{r}\right)\left[\left[N\right]\left(\bar{r},\tau \right)+Q^{2}\left[P_{2}\right]\left(\bar{r},\tau \right)\right]dV}{\int W^{2}\left(\bar{r}\right)\left[\bar{N}\left(\bar{r}\right)+Q^{2}{\bar{P}}_{2}\left(\bar{r}\right)\right]dV}$$

Here, $\bar{N}\left(\bar{r}\right)$ and $\bar{P_{2}}\left(\bar{r}\right)$ are the steady-state populations of N and P_2_ (also denoted $A_{1}\left(\bar{r}\right)$ and $A_{3}\left(\bar{r}\right)$ before), respectively. The fitting of photophysical rate parameters was then performed with a program written in Python, simulating theoretical FCS curves using Equation (10) and comparing them to the experimental data. Similar to the fitting of the experimental TRAST curves, the set of rate parameter values best describing the experimental data was then found using non-linear least squares optimization. In the fit, the excited-state lifetimes, $\tau_{f}$, of N and P_2_ were fixed to fitted values determined with TCSPC measurements (see the Results section). In the fits, the ratio of $S=\omega_{z}/\omega_{0}$ was fixed to 6.8 and $\tau_{D}$ was fitted as an individual parameter to the separate curves. 

### 4.6. Fluorescence Lifetime Measurements

Time-correlated single-photon counting (TCSPC) lifetime measurements were performed using the same experimental setup as for the FCS measurements, but now with the excitation laser (638 nm LDH-D-C-640, PicoQuant GmbH, Berlin, Germany) operated in pulsed mode. The instrument response function (IRF) was determined from the back-reflected light from the laser excitation pulses. The fluorescence signal was fed into a data acquisition card (Hydraharp, Picoquant GmbH, Berlin, Germany), deconvoluted, and then fitted to an exponential decay based on non-linear least squares minimization (Symphotime, Picoquant GmbH, Berlin, Germany).

## Figures and Tables

**Figure 1 ijms-24-01990-f001:**
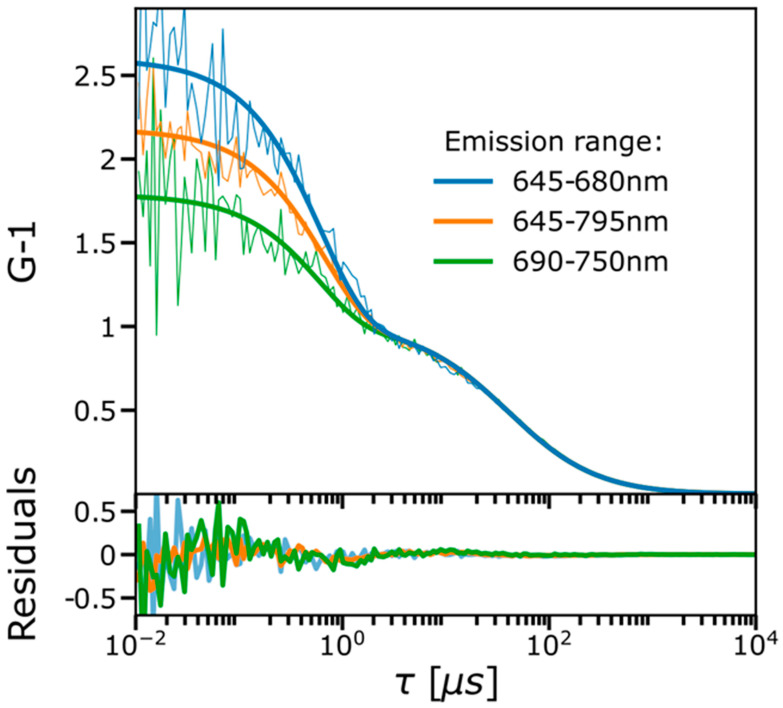
Experimental FCS curves (thin lines) recorded at 638 nm excitation (mean $\Phi_{exc}$ 21 kW/cm^2^), with the emission filter used as specified in the legend. The curves were individually fitted to a two-state model (one dark state; thick solid lines) using Equation (8). Fitted amplitudes decreased with more red-shifted emission filters used (0.61, 0.54, 0.44), while the relaxation times did not significantly change (isomerization relaxation time, $\tau_{iso}$, 0.6 µs). Fitting residuals are shown in the lower subplot.

**Figure 2 ijms-24-01990-f002:**
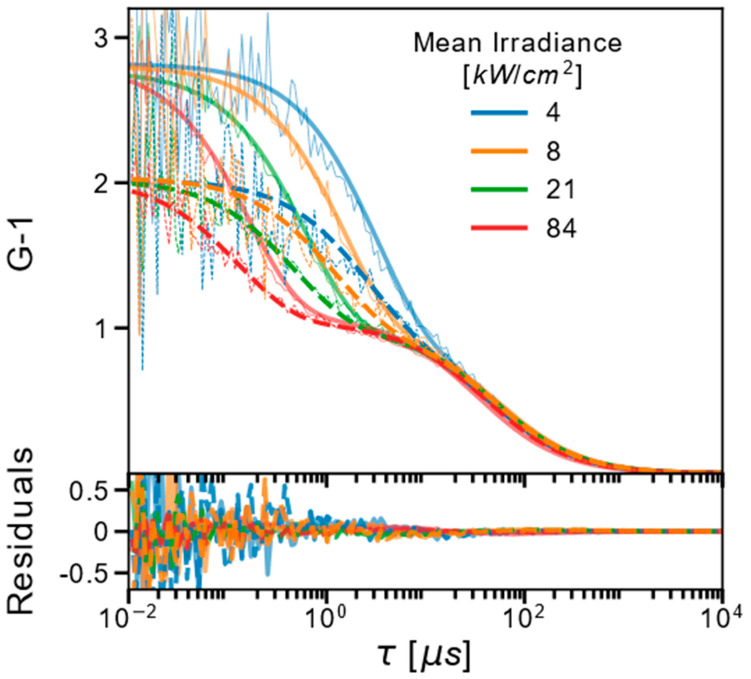
FCS curves recorded under different mean $\Phi_{exc}$ (638 nm excitation) and using two different emission filters: 645–680 nm (B-filter, data: thin solid lines; fitted curves: dim thick lines) and 690–750 nm (R-filter, data: dotted lines; fit: thick dashed lines). The FCS curves were fitted globally, as described in the main text. Fitting residuals are shown in the lower subplot.

**Figure 3 ijms-24-01990-f003:**
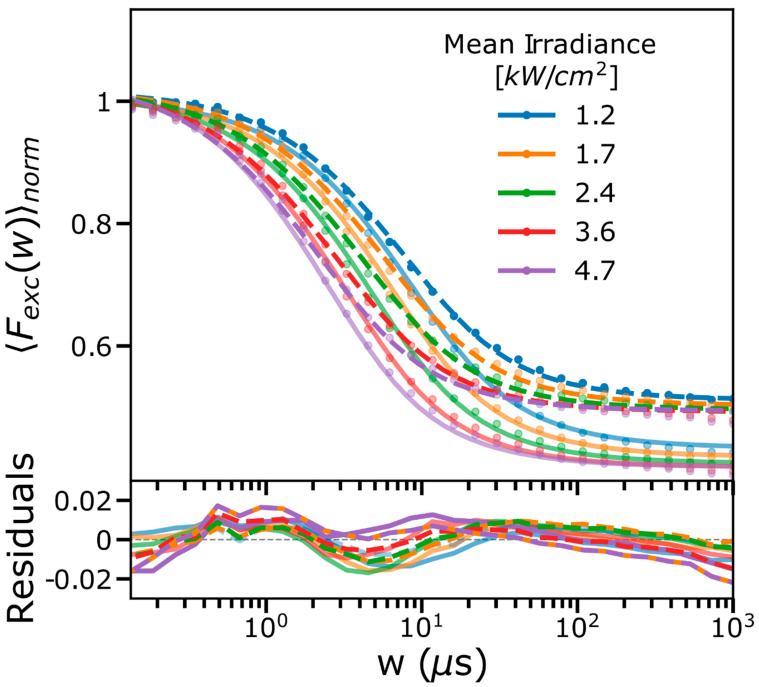
TRAST curves recorded under different mean $\Phi_{exc}$ (638 nm excitation) and using two different emission filters: 645–680 nm (B-filter, data: dim circles; fitted curves: dim thick lines) and 690–750 nm (R-filter, data: circles; fit: thick dashed lines). The TRAST curves were globally fitted, as described in the main text. Fitting residuals are shown in the lower subplot.

**Figure 4 ijms-24-01990-f004:**
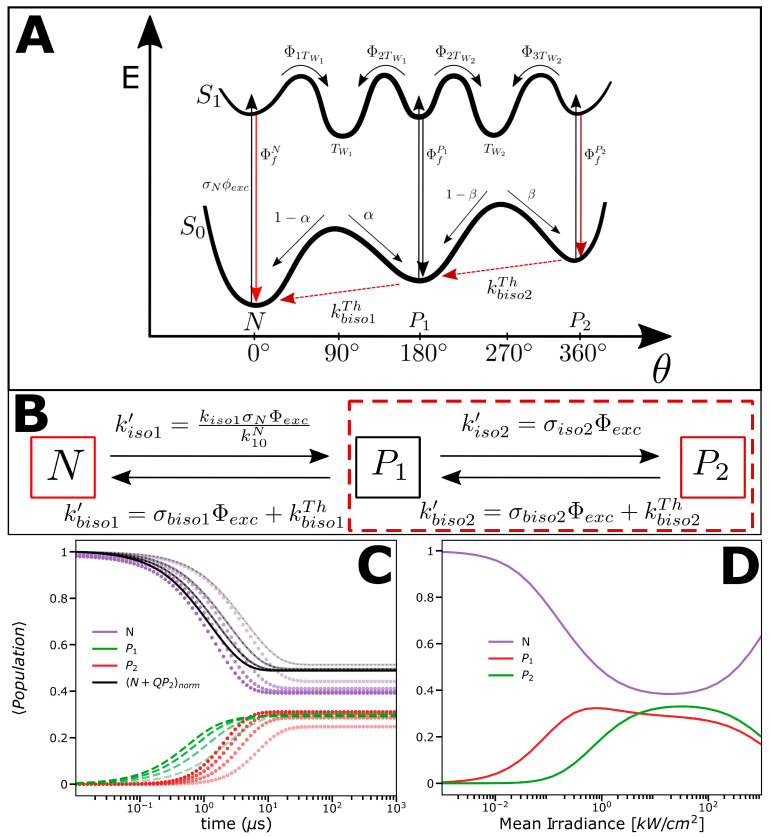
(**A**) Potential energy curve diagram for photo-isomerization of Cy5, where the energies of the ground and excited singlet states are represented as functions of the total torsion angle (Θ) and which includes an all-*trans* (N), a mono-*cis* (P_1_), and a double-*cis* (P_2_) conformation. N is excited with a rate $\sigma_{N}\Phi_{exc}$ from the singlet ground state $S_{0_{}}$ to the singlet excited state $S_{1_{}}$. The $S_{1_{}}$ state of N can decay back to the $S_{0_{}}$ state with a rate $k_{10}^{N}$, with a fluorescence quantum yield of $\Phi_{f}^{N}$, or it can transfer to a twisted intermediate state $T_{W_{1}},$ with a yield of $\Phi_{1T_{W_{1}}\;}$. The effective isomerization rate is then given by $k_{iso1}=\Phi_{1T_{W_{1}}\;}k_{10}^{N}\alpha$, where α is the branching ratio of the $T_{W_{1}}$ state to go into P_1_ (and 1- α the corresponding branching ratio to go back to N). Similarly, P_1_ is excited with a rate $\sigma_{P_{1}}\Phi_{exc}$ from its $S_{0_{}}$ state to its $S_{1_{}}$ state. The $S_{1_{}}$ state can decay back to the $S_{0_{}}$ state with a rate $k_{10}^{P_{1}}$, with a fluorescence quantum yield of $\Phi_{f}^{P_{1}}$, or it can transfer to any of the twisted intermediate states $T_{W_{1}}$ and $T_{W_{2}}$ and undergo back-isomerization to N or a second isomerization to P_2_, with rates $k_{biso1}=\Phi_{2T_{W_{1}}\;}k_{10}^{P_{1}}\left(1-\alpha \right)$ and $k_{iso2}=\Phi_{2T_{W_{2}}\;}k_{10}^{P_{1}}\left(\beta \right)$, respectively, where β is the branching ratio of the second twisted intermediate state $T_{W_{2}}$. Finally, P_2_ is excited with a rate $\sigma_{P_{2}}\Phi_{exc}$ from its $S_{0_{}}$ state to its $S_{1_{}}$ state, $k_{10}^{P_{2}}$ denotes the decay rate back to $S_{0_{}}$, and $\Phi_{f}^{P_{2}}$ is the fluorescence quantum yield of P_2_. From its $S_{1_{}}$ state, P_2_ can transfer to the twisted intermediate state $T_{W_{2}}$ and undergo back-isomerization to P_1_, with a rate $k_{biso2}=\Phi_{3T_{W_{2}}\;}k_{10}^{P_{2}}\left(1-\beta \right)$. From both P_1_ and P_2_, thermal back-isomerization from their $S_{0_{}}$ states can take place with the rates $k_{biso1}^{Th}$ and $k_{biso2}^{Th}$, respectively. In the model, and in our studies of Cy5, intersystem crossing from any of the $S_{1_{}}$ states of N, P_1_, and P_2_ to a triplet state can be neglected. Given the short (1.0 ns) excited-state lifetime of Cy5, excited singlet-state populations will be low for the $\Phi_{exc}$ applied, particularly in the TRAST experiments. Consequently, the effective rates of intersystem crossing will be low compared to typical triplet-state decay rates found in air-saturated aqueous solutions (~0.5 µs^−1^) [31,32]. Moreover, intersystem crossing will also be effectively outcompeted by the higher isomerization and back-isomerization rates of the N, P_1_, and P_2_ states. (**B**) Simplified three-state photo-isomerization model based on the model in Figure 4A and used to fit the recorded FCS and TRAST curves in Figure 2 and 3. In the TRAST and FCS experiments, the transitions between the N, P_1_, and P_2_ states occur on a timescale of µs or longer, much longer than the equilibration between the ground and excited singlet states within the N, P_1_, and P_2_ states upon onset of excitation light (their anti-bunching times), which takes place in the ns time range. Hence, in the model, we only need to consider the effective rates of isomerization and back-isomerization, with $k_{iso1}’=k_{iso1}\sigma_{N}\cdot \Phi_{exc}/k_{10}^{N}$ and $k_{iso2}’=\sigma_{iso2}\Phi_{exc}$ denoting the isomerization rates from N to P_1_ and from P_1_ to P_2_, respectively, and $k_{biso1}´=\sigma_{biso1}\Phi_{exc}+k_{biso1}^{Th}$ and $k_{biso2}´=\sigma_{biso2}\Phi_{exc}+k_{biso2}^{Th}$ denoting the back-isomerization rates from P_1_ to N and from P_2_ to P_1_, respectively. Here, $k_{iso1}$ is the isomerization rate from the excited singlet state of N to P_1_, and $\sigma_{iso2}$, $\sigma_{biso1}$, and $\sigma_{biso2}$ represent cross sections for P_1_-to-P_2_ isomerization, P_1_-to-N back-isomerization, and P_2_-to-P_1_ back-isomerization, respectively, as defined in Appendix A. $k_{biso1}^{Th}$ and $k_{biso2}^{Th}$ denote the thermal back-isomerization rates from P_1_ to N and from P_2_ to P_1_, respectively. Here, given that most cyanine dyes are in an all-*trans* (N) conformation at thermodynamic equilibrium [1,24,32], we neglect any thermal isomerization within Cy5 and thus assume that it fully returns to its N state in the absence of excitation. Finally, in the fitting of the parameter values of the model of Figure 4A to the experimental TRAST and FCS curves, the fluorescence quantum yields, as given in Figure 4A, are set to ΦfN=q1F, $\Phi_{f}^{P_{2}}=0$ and ΦfP2=q2F, i.e., with P_1_ non-emissive, and with the relative fluorescence brightness of P_2_ compared to N given by Q=(q2Fq2DσP2)/(q1Fq1DσN), Equation (2), and where *Q* also was included as a fitting parameter in the fitting of the FCS and TRAST curves in Figure 1 and Figure 2. As a possible model, we also discuss a two-state model, with P_1_ and P_2_ merged into one photo-isomerized state, P (marked with a red-dotted square in the figure). The merged P state may represent either a single, weakly fluorescent mono-*cis* state or a time average of one or several photo-isomerized forms and a double-photo-isomerized form of Cy5. See also Appendix A, and the main text for discussion. (**C**) Calculated populations of N, P_1_, and P_2_ over time after onset of excitation at 638 nm, and based on the fitted parameters, as specified in the main text. The color (see legend) indicates which state is calculated; increasing color intensities represent a higher $\Phi_{exc}$ applied. The $\Phi_{exc}$ values used in the calculations were 1.2, 2.4, 3.6, and 4.7 kW/cm^2^. Black curves represent the resulting weighted normalized fluorescence contribution from the fluorescent states N and P_2_. (**D**) Calculated steady-state populations within Cy5 upon CW excitation, based on rates and cross sections, as obtained from fitting of the experimental FCS and TRAST curves in Figure 2 and Figure 3 to the model in Figure 4B. The steady-state populations are plotted versus $\Phi_{exc}$. Colors in the legend indicate which state population is calculated. The steady-state is affected by thermal rates in the $\Phi_{exc}$ range typically used in TRAST experiments (<5 kW/cm^2^), but for the $\Phi_{exc}$ range used in the FCS experiments (≥4 kW/cm^2^), they have a small impact on the steady-state populations of N, P_1_, and P_2_.

**Figure 5 ijms-24-01990-f005:**
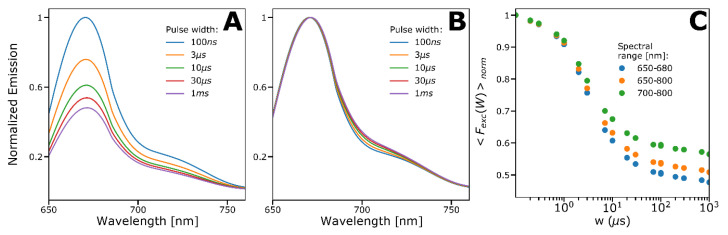
Spectral-TRAST measurements of Cy5 in PBS solution (12 mM, pH 7.2) at 638 nm excitation. The spectra in (**A**–**B**) are measured at $\Phi_{exc}$ = 3.9 kW/cm^2^ using excitation pulse trains with a constant duty cycle of 0.002 and with different pulse widths, *w*. (**A**) Emission spectra obtained for different *w* (specified in legend), normalized with the emission maximum retrieved at 100 ns. (**B**) Emission spectra obtained for different *w* (specified in legend), normalized with the emission maximum for each curve. (**C**) TRAST curves generated from fluorescence within different spectral windows of the spectra in (**A**) and with the different spectral windows specified in the legend. The TRAST curves were normalized so that the fluorescence intensity recorded with *w* = 100 ns within the different spectral windows (blue curve in (**A**)) was set to unity.

## Data Availability

All raw data on which this report is based can be assessed (17 January 2023) at https://doi.org/10.5281/zenodo.7405074.

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
