# Peer review of "Combined Fluorescence Fluctuation and Spectrofluorometric Measurements Reveal a Red-Shifted, Near-IR Emissive Photo-Isomerized Form of Cyanine 5"

_ijms, 2023, doi:10.3390/ijms24031990_

Round 1

Reviewer 1 Report

The authors consider in detail the primary photophysical and photochemical processes in the molecule of the cyanine dye Cyanine 5. With the help of subtle experiments (FCS and TRAST methods) with registration of fluorescence in different spectral ranges, it was shown that an additional fluorescent state of the dye is involved in deactivation processes of electronic excitation. A three-state model of electronic levels is proposed and considered in detail mathematically. The model is equivalent to that proposed earlier for DTDCI (P. Vaveliuk, L.B. Scaffardi, R. Duchowicz. J. Phys. Chem. 1996, 100, 11630-11635). The article may be published in the Journal after some revision.

1. The authors ignore the role of the triplet state in the process: it is assumed that the intersystem crossing to the triplet state is zero. But in the work (Z. Huang, D. Ji, A. Xia, F. Koberling, M. Patting, R. Erdmann. J. Am. Chem. Soc. 2005, 127, 8064-8066) it was shown that the triplet state can play a role in (delayed) fluorescence of Cyanine 5.

2. In Supplementary, in the caption to Fig. S2A the lifetime as a result of fitting should be specified. In the caption to Fig. S2B “B” should be replaced by “A”.

Author Response

First, we would like to thank Reviewer 1 for the useful comments.

  1. The authors ignore the role of the triplet state in the process: it is assumed that the intersystem crossing to the triplet state is zero. But in the work (Z. Huang, D. Ji, A. Xia, F. Koberling, M. Patting, R. Erdmann. J. Am. Chem. Soc. 2005, 127, 8064-8066) it was shown that the triplet state can play a role in (delayed) fluorescence of Cyanine 5.

We agree that the article from Huang et al from 2005 is relevant for our work. We have referred to another article by the same group and on a similar topic (ref 9, from 2006), but have now also added this paper into the reference list (as ref 39). In the figure caption of Figure 4A, we argued that at the excitation intensities applied, the effective rates of intersystem crossing are low, compared to typical triplet state decay rates found in air-saturated aqueous solutions (~0.5µs-1). Moreover, intersystem crossing will also be effectively outcompeted by the higher isomerization and back-isomerization rates of the N, P1 and P2 states. Thus, triplet state formation can be disregarded. The experiments in refs 9 and 39 were done in deoxygenated ethanol, where triplet decay rates will be orders of magnitude lower than in air-saturated ethanol, or in air-saturated water. Therefore, even if the triplet formation rates are low, triplet state population buildup can under such conditions still be clearly detected.  To add clarity and explain why triplet state formation can be disregarded under our conditions, while clearly seen in the work of ref 9 and 39, we have added a short section in the discussion part (lines 360-367).

  1. In Supplementary, in the caption to Fig. S2A the lifetime as a result of fitting should be specified. In the caption to Fig. S2B “B” should be replaced by “A”.

The result of the fitting has been added in Fig. S2A. We thank the reviewer for finding this typo. “B” has now been replaced by “A” in the figure caption of Fig S2B.

A few additional minor corrections have been made in the main text, in particular:

  • Insets of Figure 1: We found that the insets in Figure 1 were in the wrong order. Blue line now correctly indicates 690-750nm, and green line 645-680nm.
  • Figure 4A: The notations for the fluorescence quantum yields of N, P1 and P2 in the model figure have been corrected.

All corrections are marked with via the “track changes” function.  

Reviewer 2 Report

Dear Editor,

            This manuscript investigated the photo-isomerization of the pentamethine Cyanine 5 (Cy5) fluorescence correlation spectroscopy (FCS) and transient state (TRAST) excitation-modulation spectroscopy. Have the following suggestions and comments to improve the state of the study. So, I recommend accept after some minor revisions.

Minor points: Line 12: continuous exciation,= continuous excitation, Line 12: population probabilies= population probabilities; Line 32: but has over=but over; Line 50: decades been boosted = decades have been boosted; line 253: brightness= brightness; Line 300:  fitted values= fitted values; Line 421:  edge fiter= edge filter.

Line 484: What does "adius" mean?

Line 577: The authors should add the references for the equations mentioned here or in the Suplementary Material.

The authors could insert after the Discussion section a conclusion section including numerical data.

Author Response

First, we would like to thank the Reviewer 2 for the useful comments.

Minor points: Line 12: continuous exciation,= continuous excitation, Line 12: population probabilies= population probabilities; Line 32: but has over=but over; Line 50: decades been boosted = decades have been boosted; line 253: brightness= brightness; Line 300:  fitted values= fitted values; Line 421:  edge fiter= edge filter.

We thank the reviewer for finding these misprints. They have now been corrected.

Line 484: What does "adius" mean?

Should be “radius”. It is now corrected.

Line 577: The authors should add the references for the equations mentioned here or in the Suplementary Material.

For Eqs S1-S8 a reference to ref 32 was added in the beginning of Section S1. References to the equation S9 (Section S2) has been added in the main text (line 570). Additionally, references to the equations were added on line 575 and line 577.

Additionally,

The authors could insert after the Discussion section a conclusion section including numerical data.

A brief conclusion section has now been added at the end of the Discussion section (lines 401-416), were we state the obtained isomerization rate parameter value obtained and summarize the main findings.

A few additional minor corrections have been made in the main text, in particular:

  • Insets of Figure 1: We found that the insets in Figure 1 were in the wrong order. Blue line now correctly indicates 690-750nm, and green line 645-680nm.
  • Figure 4A: The notations for the fluorescence quantum yields of N, P1 and P2 in the model figure have been corrected.

All corrections are marked with via the “track changes” function.  
